# *Myosin 1f* and *Proline-rich 13* are transcriptionally upregulated yet functionally redundant in CD4+ T cells during blood-stage *Plasmodium* infection

**Takahiro Asatsuma, Marcela L. Moreira, Hyun J. Lee, Brooke J. Wanrooy, Oliver P. Skinner, Shihan Li, Ivana Rea, Taidhgin Harkin, Saba Asad, Cameron G. Williams, Lynette Beattie, Ashraful Haque**◯*

Department of Microbiology and Immunology, University of Melbourne, The Peter Doherty Institute for Infection and Immunity, Parkville, Victoria, Australia

* ashraful.haque@unimelb.edu.au

## Abstract

*Plasmodium*-specific CD4+ T cells differentiate into effector and memory subsets during experimental malaria, via mechanisms that remain incompletely characterised. By mining scRNA-seq data of CD4+ T cells during *Plasmodium chabaudi chabaudi* AS infection in mice, we identified two genes previously uncharacterised in T helper cells, long-tailed unconventional myosin 1f (*Myo1f*) and proline-rich13/taxanes-resistance 1 (*Prr13/Txr1*), which were upregulated during effector and memory differentiation. Myo1f is reported to regulate motility and granule exocytosis in myeloid and γδ T cells. Prr13/Txr1 is reported to transcriptionally regulate sensitivity to anti-cancer drugs. To test for cell-intrinsic gene function, we generated *Plasmodium*-specific TCR transgenic, PbTII cells harbouring CD4-promoter driven Cre recombinase and target genes with *loxP*-flanked essential exons. We validated our approach for the transcription factor *Maf*, formally demonstrating here that cMaf is essential for T follicular helper (Tfh) cell differentiation in experimental malaria. Next, having generated conditional knockout lines for *Myo1f* and *Prr13,* we observed that deficiency in *Myo1f* or *Prr13* had no impact on either clonal expansion, Th1/Tfh differentiation or transit to memory. Additionally, despite continued expression during re-infection, *Myo1f* was unnecessary for Th1 recall *in vivo*. Thus, while cMaf is critical for Tfh differentiation in experimental malaria, *Myo1f* and *Prr13*, although transcriptionally upregulated, are unnecessary for effector or memory CD4+ T cell responses.

## Introduction

Malaria continues to threaten global human health, with approximately 249 million cases and 608,000 deaths worldwide in 2022 [1]. During experimental blood-stage *Plasmodium* infection, antigen-specific CD4+ T cells orchestrate cellular and humoral immune responses through various T helper (Th) cell types [2,3]. T helper 1 (Th1) cells are protective likely due

**Data availability statement:** All relevant data are within the paper and Supporting information files.

**Funding:** Ashraful Haque received funding for this study from the Australian National Health & Medical Research Council. grant numbers: 1126399 & 1180951 https://www.nhmrc.gov.au/. The funders played no role in the study design, data collection and analysis, decision to publish, or preparation of the manuscript.

**Competing interests:** The authors have declared that no competing interests exist.

to production of pro-inflammatory cytokines interferon gamma (IFN-γ) and tumor necrosis factor alpha (TNF-α), which may limit parasite growth via phagocyte activation [4,5]. T-follicular helper (Tfh) cells support the production of high-affinity *Plasmodium*-specific antibodies in germinal centres [6,7]. Regulation of inflammatory responses can occur through IL-10-producing Type I regulatory T cells (Tr1), although this response can also restrict parasite-killing [8–12]. Thus, protective immunity to malaria relies on a balanced response by effector CD4+ T cells.

Discovery of new genes that control CD4+ T cell responses may provide opportunities to improve immunity to malaria. Advancements in single-cell RNA sequencing (scRNA-seq) and utilization of TCR-transgenic CD4+ T (PbTII) cells specific for *Plasmodium* Heat Shock Protein 90 previously enabled us to map transcriptional dynamics accompanying CD4+ effector T cell differentiation, transition to memory, and recall during re-infection in mice [11,13–15]. For instance, *Maf*, which promotes Tfh cell differentiation *in vivo* during immunization [16,17], is up-regulated during both Th1 and Tfh cell differentiation in experimental malaria. In this study, based on our existing scRNA-seq data [11,13–15], we focussed on two genes previously unstudied in CD4+ T cells: long-tailed unconventional myosin 1f (*Myo1f*) and proline-rich 13/taxanes-resistance 1 (*Prr13/Txr1*).

Myosins are actin-dependent motor proteins, comprising at least 18 distinct classes [18,19]. Class I myosins are the largest group, with six genes encoding short-tailed forms (Myosin 1a, -b, -c, -d, -g, and -h) and two encoding unconventional long-tailed forms (Myo1e and Myo1f), in mice and humans [20]. Recent studies report roles for Myo1e and Myo1f, in various cellular processes such as migration, adhesion, endocytosis, exocytosis, and phagocytosis [20–22]. Myo1f expression occurs in many tissues including spleen, lymph nodes, thymus, and lung, particularly by natural killer (NK) cells, neutrophils, macrophages, and dendritic cells within lymphoid tissues [23]. Myo1f-deficiency impacted the adhesion and migration of neutrophils, mast cells, and γδT intraepithelial lymphocytes [23–26]. Additionally, Myo1f was reported to contribute to several cellular processes, including granule exocytosis in neutrophils, IgE-dependent mast cell degranulation, and macrophage-mediated phagocytosis [23,27,28]. These processes, along with the actions of Myo1e, are regulated by controlling actin dynamics at the cell membrane [28,29]. These studies highlight that Myo1f plays a crucial role in innate immune cell migration and function. However, potential roles in adaptive immune cells remain largely unexplored.

*Prr13/Txr1* encodes for a nuclear protein and likely transcriptional regulators, whose up-regulation in cancer is reported to promote resistance to chemotherapy drugs known as taxanes [30–34]. *Prr13/Txr1* appears to induce taxane resistance through its association with thrombospondin-1 (TSP-1/THBS-1) and CD47 within cells [34,35]. It has been shown that *Prr13* upregulation in certain cancer cells leads to a significant downregulation of TSP-1, a protein known for its anti-angiogenic and pro-apoptotic properties, which binds to cell surface receptor CD47 and triggering caspase-independent apoptosis of cancer cells [34,36,37]. This inverse relationship between *Prr13* and TSP-1 suggests that high levels of *Prr13* reduce the apoptotic response to taxane treatments by decreasing TSP-1 levels. A role for *Prr13* in adaptive immune cells has yet to be examined.

In this study, we utilised existing scRNA-seq data from a mouse model of blood-stage malaria to hypothesise possible roles for *Myo1f* and *Prr13* in CD4+ T cells. We next established a method to test for T-cell intrinsic gene function in *Plasmodium*-specific TCR transgenic CD4+ T cells during infection, using the transcription factor, *Maf*, as a test case. Finally, we tested whether *Myo1f* or *Prr13* are required during effector or memory CD4+ T cell responses *in vivo*.

## Results

### *Myo1f* and *Prr13* are transcriptionally upregulated in *Plasmodium*-specific CD4⁺ T cells during experimental malaria

Firstly, we mined existing scRNA-seq datasets of splenic TCR transgenic PbTIIs, and CD11a$^{hi}$ CXCR3⁺ polyclonal CD4⁺ T cells during primary and secondary *P. chabaudi chabaudi* AS (*Pc*AS) infection in mice. We aimed to identify novel genes whose expression dynamics suggested possible function in effector and/or memory CD4⁺ T cells [11,14,38]. Here, we observed that *Myo1f* was not expressed in naïve PbTIIs, but was upregulated during Th1 effector and memory differentiation, less so for Tfh-differentiation (Fig 1Ai-iii). This was corroborated by assessment of polyclonal CD11a$^{hi}$ CXCR3⁺ CD4⁺ T cells at 7-days post-infection (Fig 1Bi-ii). Given prior evidence of Myo1f's role in regulating movement and granule exocytosis in innate immune cells, particularly neutrophils [21,24–27], we hypothesized that this myosin protein might also control T cell migration and function during infection. *Prr13* was modestly expressed in naïve PbTIIs, and strongly upregulated by all PbTIIs and polyclonal counter parts (Fig 1Ai-iii and 1Bi-ii), with the magnitude and frequency of expression largely maintained by PbTIIs until day 28 post-infection and treatment (Fig 1Aii-iii). After re-infection, expression of *Myo1f* and *Prr13* was retained, without evidence of further up-regulation in PbTIIs (Fig 1Ci-ii) or polyclonal cells (Fig 1Di-ii). Given sequence predictions and prior studies suggesting that Prr13 may encode a DNA-binding protein localized in the nucleus and function as a transcriptional regulator [31,33–35], we surmised that this protein could act as a transcription factor in T cells, controlling differentiation and cellular maintenance processes. Taken together, our analysis revealed *Myo1f* and *Prr13* expression were upregulated and maintained in specific effector and memory CD4⁺ T cell subsets during experimental malaria.

### An *in vivo* approach for testing gene function in *Plasmodium*-specific CD4⁺ T cells

To test for cell-intrinsic gene function in *Plasmodium*-specific CD4⁺ T cells, we developed an adoptive transfer system. In this system, donor PbTIIs were obtained either from knockout (KO) mouse, which constitutively express Cre-recombinase (driven by the CD4-promoter) and harbor homozygous alleles for a target gene with essential exons flanked by *loxP* sites (denoted as $^{(f/f)}$ here), or from wild-type (WT) mouse with the $^{(f/f)}$ alleles but without Cre expression. These donor cells were then adoptively transferred into congenic recipient mice infected with *Pc*AS. We tested our approach on *Maf*, a gene that promotes Tfh-differentiation during immunization [16,17], and IL-10 production in malaria [12,39], but whose role in Tfh-differentiation during *Plasmodium* infection remains to be formally demonstrated. We examined PbTII cells at 7-days post-infection, the peak of parasitemia, when Th1 and Tfh differentiation was readily detectable, as previously demonstrated [14].

Firstly, our scRNA-seq data confirmed strong *Maf* upregulation by PbTII Th1 and Tfh cells during *Pc*AS infection (Fig 2Ai-ii). We transferred into congenic recipients either CD4-Cre *Maf* $^{(f/f)}$ PbTIIs (KO), or WT control PbTIIs from *Maf* $^{(f/f)}$ PbTII littermates, examining splenic responses 7-days post-infection (Fig 2B). Firstly, we observed no up-regulation of cMaf in *Maf*-deficient PbTIIs, confirming this method of gene depletion using our adoptive transfer system (Fig 2C and D). PbTIIs *Maf*-deficient PbTIIs displayed no defect in clonal expansion compared to wild-type controls (Fig 2E), nor in the upregulation of the Th1-associated chemokine receptor CXCR6 (Fig 2F). In stark contrast, *Maf*-deficient PbTIIs exhibited a profound defect in upregulation of Tfh-associated chemokine receptor, CXCR5, compared to *Maf*-sufficient controls (Fig 2Fi-ii). This defect was absent as expected amongst WT endogenous polyclonal CD4⁺ T cells in recipient mice (Fig 2Fi-ii), confirming a cell-intrinsic

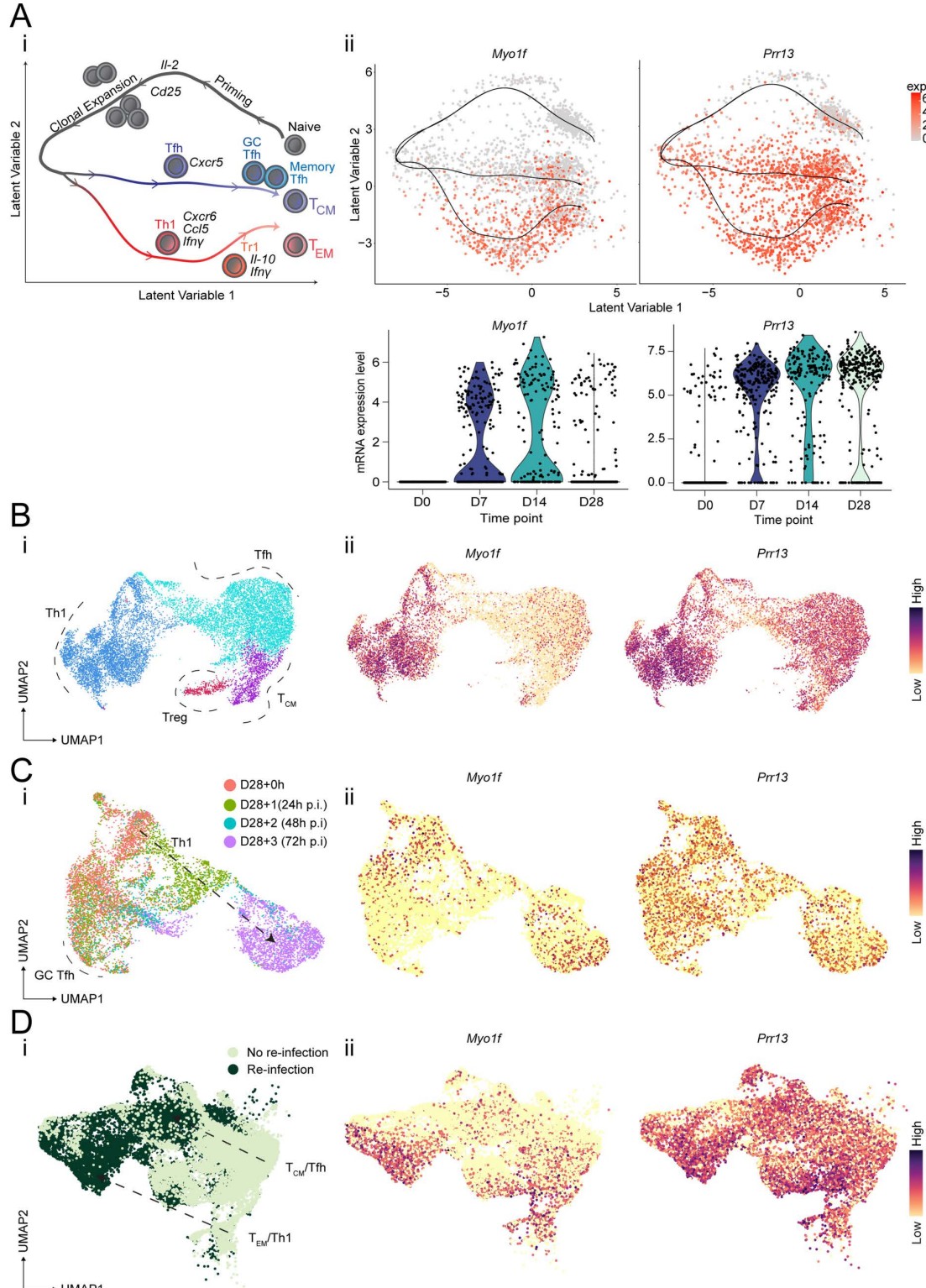

**Fig 1.** *Myo1f* **and** *Prr13* **are transcriptionally upregulated in** *Plasmodium***-specific CD4** [+] **T cells during experimental malaria.** (Ai) Trajectories and gene-expression dynamics of PbTII cells during blood-stage malaria infection in mice. (Aii & Aiii) UMAP and Violin plot of PbTII cells expressing *Myo1f* and *Prr13* at 0, 7, 14, and 28 d.p.i. UMAP obtained from Graphical User Interface (https://haquelab.mdhs.unimelb.edu.au/cd4_memory/). (Bi & Bii) UMAP of polyclonal CD4+ T cells expressing *Myo1f* and *Prr13* at 7 d.p.i. obtained from the study by Williams et al [38]. (**Ci & Cii**) UMAP of antigen-experienced PbTII

expressing *Myo1f* and *Prr13* prior to (D28 + 0) and 24h, 48h, and 72h (D28 + 1, D28 + 2, and D28 + 3, respectively) after *Plasmodium* re-infection. (**Di & Dii**) UMAP of polyclonal CD4$^+$ T cells expressing *Myo1f* and *Prr13* prior to and 3 days after re-infection at 28 d.p.i. (**C & D**) UMAP obtained from Graphical User Interface (https://haquelab.mdhs.unimelb.edu.au/cd4_re-infection/). (**A-D**) Reprinted from [11].

requirement for *Maf* in promoting CXCR5 expression in PbTIIs. Similar patterns were observed for intracellular expression of Th1/Tfh-associated master transcription factors, Tbet and Bcl6, with *Maf* being essential for Bcl6 upregulation but not Tbet expression (Fig 2Gi-ii). Collectively, these data demonstrated, firstly, the utility of our approach for examining cell-intrinsic gene function in *Plasmodium*-specific CD4$^+$ T cells *in vivo*, and secondly that cMaf is essential for Tfh differentiation in experimental malaria.

### Generation of *Myo1f*- and *Prr13*-deficient mice

Next, we generated mice in which *Myo1f* or *Prr13* were deleted in CD4$^+$ TCR transgenic (PbTII) cells (Fig 3A and B). C57BL/6 mice expressing Cre recombinase under the control of the CD4 promoter were crossed with a strain harbouring loxP sites flanking exon 2 and 4 of the *Myo1f* gene or flanking exon 3 and 4 of the *Prr13* gene (Fig 3A). In parallel, gene-targeted strains were also crossed with PbTII TCR-transgenic mice (Fig 3B). Offspring was further mated to generate *Myo1f*-deficient (CD4-Cre *Myo1f*$^{(f/f)}$) PbTII mice, or *Prr13*-deficient (CD4-Cre *Prr13*$^{(f/f)}$) PbTII mice. Expression of *loxP* sites and Cre-mediated excision was confirmed by polymerase chain reaction (PCR) (Fig 3Ci-iii). *Myo1f*-sufficient (*Myo1f*$^{(f/f)}$) or *Prr13*-sufficient (*Prr13*$^{(f/f)}$) PbTII littermates mice were used as WT PbTII donors in this study (Fig 3A-C).

### *Prr13* is not required for effector or memory CD4$^+$ T cell responses during experimental malaria

As a DNA-binding protein with transcriptional regulatory potential [30–35], we next hypothesized *Prr13* promoted Th1/Tfh differentiation and transition to memory during *Pc*AS infection. To test this, we transferred *Prr13*-deficient (CD4-Cre *Prr13*$^{(f/f)}$) or *Prr13*-sufficient (*Prr13*$^{(f/f)}$) PbTII cells into congenic recipient mice, and assessed splenic and hepatic responses at 7- and 28-days post-infection with anti-malarial drug treatment from day 7 p.i. to better facilitate memory transition over T-cell exhaustion (Fig 4A). *Prr13*-deficiency exerted no effect on the number of PbTIIs recovered from spleen or liver at either timepoint (Fig 4Bi-ii), nor in their expression of Th1/Tfh-associated chemokine receptors, CXCR6 and CXCR5 (Fig 4Ci-ii). Moreover, *Prr13* was not required either for direct *ex vivo* expression of Th1-associated IFNγ or CCL5, or an *in vitro* restimulated Tr1 phenotype, assessed by co-expression of IFNγ and IL-10 (Fig 4Di-ii). Taken together, our data indicated that *Prr13* is not necessary for Th1/Tfh/Tr1 differentiation during *Pc*AS infection. By 4-weeks post-infection and treatment, PbTII cells and polyclonal CD4$^+$ T cells exhibit GC Tfh (PD1$^{hi}$ CXCR5$^{hi}$), Tfh/T$_{CM}$-like (CXCR5$^+$ PD1$^{lo}$) or Th1-T$_{EM}$ (CXCR5$^{lo}$ CXCR6$^+$) memory phenotypes in the spleen (Fig 1A) [11,15]. We noted *Prr13* was required neither for the development of any of these cell-types (Fig 5Ai-ii), nor their capacity in the spleen or liver to produce IFNγ upon *in vitro* re-stimulation (Fig 5Bi-ii). These findings indicated that despite clear transcriptional up-regulation, *Prr13* was not required for any aspect of the *Plasmodium*-specific CD4$^+$ T cell response in this study.

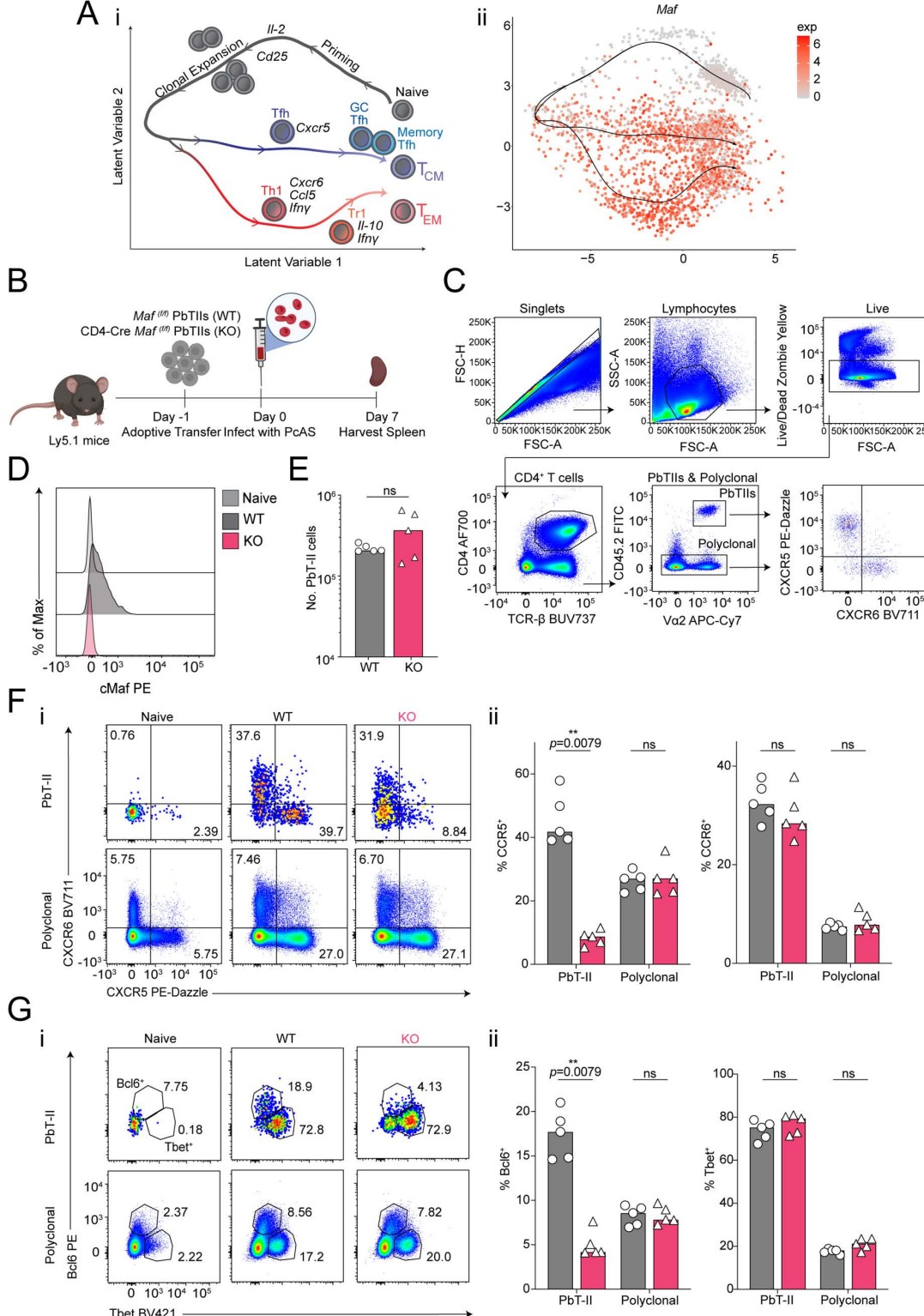

**Fig 2. An *in vivo* approach for testing gene function in *Plasmodium*-specific CD4 + T cells.** (**Ai**) Trajectories and gene-expression dynamics of PbTII cells during blood-stage malaria infection in mice. (**Aii**) UMAP of PbTII cells expressing *Maf* over the corse of infection. UMAP obtained from Graphical User Interface (https://haquelab.mdhs.unimelb.edu.au/cd4_memory/). (B) Schematic of experiment: Congenically marked (CD45.2+) *Maf* (f/f) wild-type PbTIIs (WT) or CD4-Cre *Maf* (f/f) PbTIIs (KO) were adoptively transferred into Ly5.1 (CD45.1+) recipient mice infected with *Pc*AS infection a day 0. Splenocytes

were analysed on day 7 p.i. Created with Biorender.com. (C) The gating strategy for congenically marked (CD45.2$^+$) *Plasmodium*-specific CD4+ TCR transgenic (PbTII) in Ly5.1 (CD45.2$^-$) recipient mouse. Representative FACS plots of PbTIIs (CD45.2$^+$ Va2$^+$CD4$^+$TCRβ$^+$) and polyclonal CD4$^+$ T cells (CD45.2$^-$ CD4$^+$TCRβ$^+$) gated on Live/Lymphocytes/Singlets. (D) Histogram of cMaf expression in Naïve, WT, and KO PbTIIs at day 7 p.i. **n** = 5 per group, data are generated from one experiment. (E) Total number of PbTIIs per spleen at day 7 p.i. (Fi) Representative FACS plots of CXCR5 and CXCR6 expression. (**Fii**) Bar graphs showing frequencies of CXCR5$^+$ and/or CXCR6$^+$ PbTIIs and polyclonal CD4$^+$ T cells in the WT or KO group in the spleen at day 7 p.i. (Gi) Representative FASC plots of direct *ex vivo* Bcl6 and Tbet expression. (Gii) Bar graphs showing frequencies of Bcl6$^+$ or Tbet$^+$ PbTIIs and polyclonal CD4$^+$ T cells in WT or KO group in the spleen at day 7 p.i. Cells were treated with Brefeldin A (BFA) for 3 hours at 37°C, 5% CO$_2$. (E-G) n = 5 per group, data are representative of two independent experiments. Statistical tests performed using unpaired, non-parametric Mann-Whitney t test; **$p < 0.01$.

### *Myo1f* is not required for effector or memory CD4$^+$ T cell responses during experimental malaria

Although Myo1f has been studied in innate immune cells [23–28], its function in conventional T cells remains unknown. Given its strong transcriptional upregulation in Th1 cells during both primary and secondary *Pc*AS infection (Fig 1A-D), we next hypothesized that *Myo1f* controlled Th1 differentiation. To test this, congenic mice received *Myo1f*-deficient (CD4-Cre *Myo1f* $^{(f/f)}$) or *Myo1f*-sufficient (*Myo1f* $^{(f/f)}$) PbTII cells, were infected with *Pc*AS, and assessed 7 days later for splenic and hepatic responses (Fig 6A). As for *Prr13*, we noted that *Myo1f* was not required for clonal expansion of PbTIIs (Fig 6Bi-ii), nor the upregulation of CXCR5/CXCR6 (Fig 6Ci-ii), nor direct *ex vivo* expression of Th1-associated IFNγ or Tbet (Fig 6Di-ii). Furthermore, *Myo1f* was not essential for *in vitro* Tr1 responses in the spleen or liver (Fig 6Ei-ii). Finally, given evidence *Myo1f* expression during secondary infection (Fig 1C and D), we tested for a role for this gene in Th1-recall 3-days after re-infection (Fig 7A). Although, PbTIIs had increased in number in the spleen, there was no evidence of a requirement for *Myo1f* in this process in spleen or liver (Fig 7Bi-ii). Similarly, upregulation of IFNγ or CXCR6 upon re-infection proceeded normally in the absence of *Myo1f* (Fig 7Ci-ii). Together, our data demonstrated that *Myo1f* is not required for Th1 differentiation or memory recall in experimental malaria.

## Discussion

In this study we mined existing scRNA-seq datasets derived from *Plasmodium*-specific TCR transgenic and polyclonal CD4$^+$ T cells during experimental malaria for novel genes that might control effector and memory differentiation. *Myo1f* was chosen for further investigation due to its reported role in controlling motility and granule exocytosis in innate immune cells [23–28]. *Prr13* was prioritized based on its reported role as transcriptional regulator [30–35]. To test these, we generated cell-specific gene-targeted mice and examined their gene function *in vivo*. Conclusions from our experiments are that neither of these genes, though strongly upregulated transcriptionally at various phases of effector and memory differentiation, play any role in CD4$^+$ T cells during infection with blood-stage *Plasmodium* parasites. This contrasts with *Maf*, which we formally demonstrate here plays a critical role in Tfh differentiation in experimental malaria, consistent with observations in other experimental models.

In interpreting our negative results, a central issue is whether our gene targeting approach for *Myo1f* and *Prr13* was performed correctly, since we were unable to screen for evidence of protein expression. PCR of genomic DNA from CD4-Cre *Myo1f* $^{(f/f)}$ or CD4-Cre *Prr13* $^{(f/f)}$ indicated that exon excision had successfully occurred. However, whether unanticipated, truncated versions of these proteins retained some function remains untested. A further limitation of this study is that WT and KO PbTII cells could not be co-transferred into the same

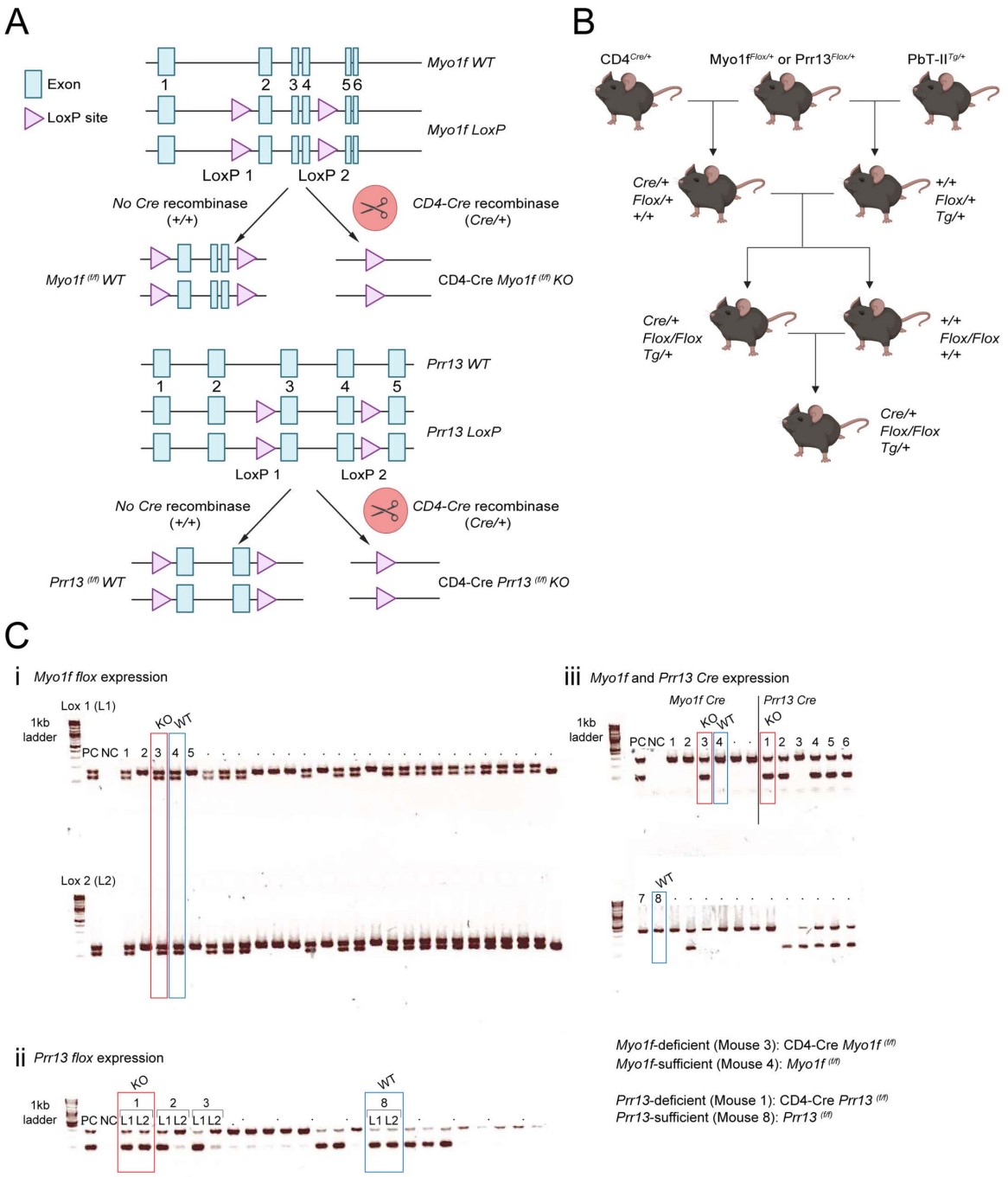

**Fig 3. Generation of *Myo1f*- and *Prr13*-deficient mice.** (A) Scheme of the floxed *Myo1f* and *Prr13* locus with *Cre* recombination. (B) The breeding strategy of generation of target-gene floxed (*Flox/Fox*) mouse expressing Cre (*Cre/+*) and *Plasmodium* antigen-specific transgene (*Tg/+*). (A & B) Created with Biorender.com. (**Ci-iii**) Intact gel bands and confirmation of expression of *flox* and *Cre* by polymerase chain reaction (PCR). *Myo1f*-sufficient (*Myo1f* (f/f)) PbTII, *Myo1f*-deficient (CD4-Cre *Myo1f* (f/f)) PbTII, *Prr13*-sufficient (*Prr13* (f/f)) PbTII, and *Prr13*-deficient (CD4-Cre *Prr13* (f/f)) PbTII mouse were used in the study. The expression of Lox 1 (L1), Lox 2 (L2), and Cre recombinase in the individual *Myo1f* or *Prr13* conditional knockout mouse. (Ci & Ciii) The mouse 3 as *Myo1f*-deficient (CD4-Cre *Myo1f* (f/f)) PbTII mouse and mouse 4 as *Myo1f*-sufficient (*Myo1f* (f/f)) PbTII mouse were used in the study. (Cii & Ciii) The mouse 1 as *Prr13*-deficient (CD4-Cre *Prr13* (f/f)) PbTII mouse and mouse 8 as *Prr13*-sufficient (*Prr13* (f/f)) PbTII were used in the study. Positive control (PC) and negative control (NC) were also assessed together. The original gel results are provided in the Supporting information.

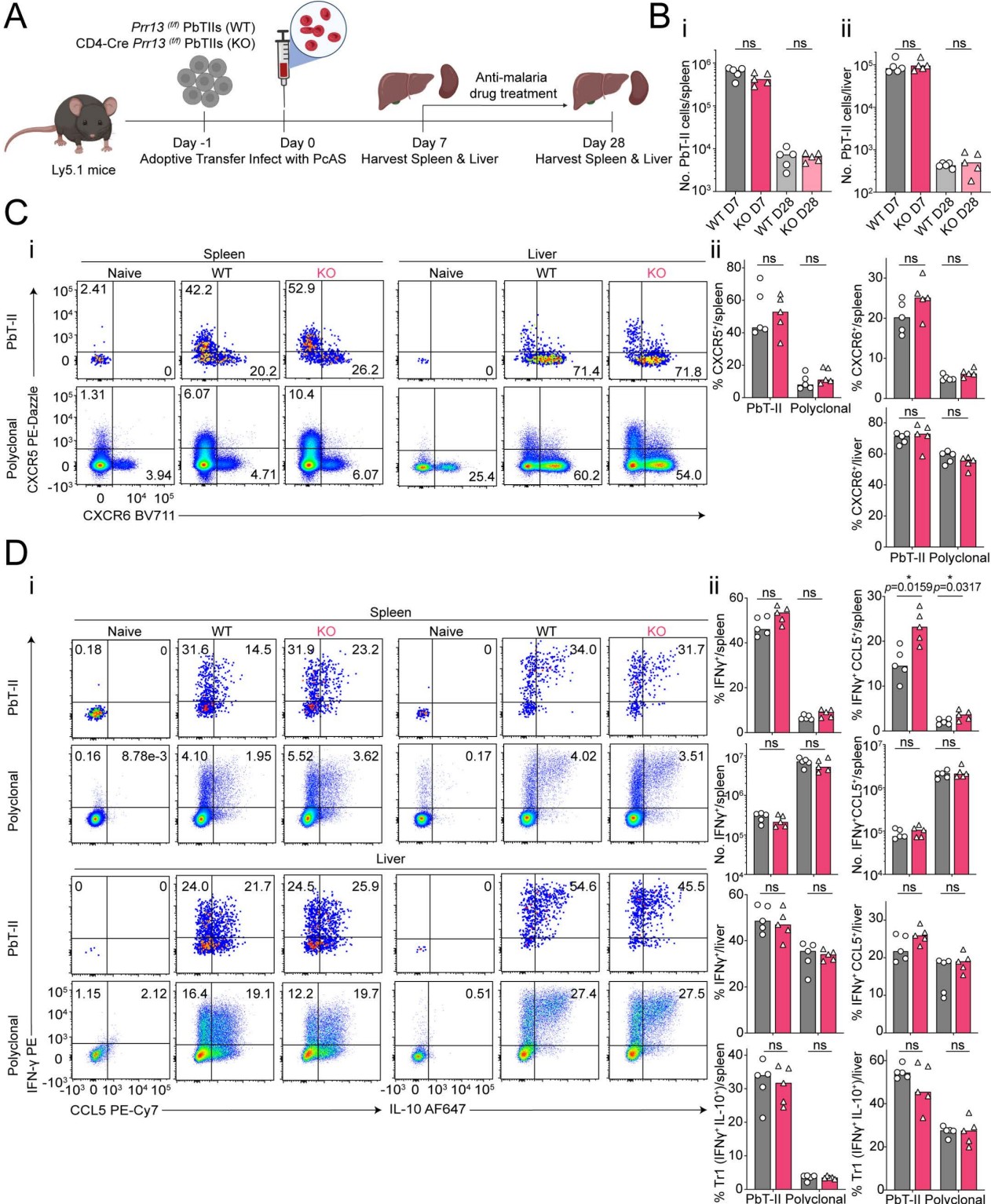

**Fig 4. *Prr13* is not required for effector CD4 + T cell responses during experimental malaria.** (A) Schematic of experiment: congenically marked (CD45.2+) *Prr13* (f/f) wild-type PbTIIs (WT) or CD4-Cre *Prr13* (f/f) PbTIIs (KO) were adoptively transferred into Ly5.1 (CD45.1+) recipient mice at day -1 and infected with *Pc*AS infection at day 0. Splenocytes and hepatocytes were analysed on day 7 and day 28 p.i. Mice underwent anti-malarial drug treatment from day 7 p.i. Created with Biorender.com. (Bi & Bii) Total number of PbTIIs per spleen and liver at day 7 and day 28 p.i. (**Ci**) Representative FACS plots of CXCR5 and CXCR6 expression. (Cii) Bar graphs showing frequencies of CXCR5+ and/or CXCR6+ PbTII

and polyclonal CD4⁺ T cells in the WT and group in the spleen and liver at day 7 p.i. (**Di**) Representative FACS plots of direct *ex vivo* IFN-γ and CCL5 expression and of *ex vivo* stimulation-induced IFN-γ and IL-10 expression. (Dii) Bar graphs showing frequencies and/or the total number of IFN-γ⁺, IFN-γ⁺ CCL5⁺, or IFN-γ⁺ IL-10⁺ Tr1 PbTII and polyclonal CD4⁺ T cells in WT and KO PbTII cells in the spleen and liver at day 7 p.i. Direct *ex vivo* was performed with intracellular staining with Brefeldin A (BFA), whereas *ex vivo* stimulation was conducted with the intracellular staining with BFA, PMA and ionomycin. (B-D) *n* = 5 per group, data are representative of two independent experiments. Statistical tests performed using unpaired, non-parametric Mann-Whitney t test; *$p < 0.05$.

recipient mouse, as both cell types expressed CD45.2. Experiments were thus performed in separate small cohorts, which reduced statistical power. Future studies using congenic markers could enable direct comparisons within greater numbers of the same recipient mice.

Targeting *Maf* served two purposes in our study. Firstly, it allowed us to validate our approach for *Plasmodium*-specific CD4⁺ T cells. The approach was powerful since all other components of the immune system were left intact, allowing us to test for cell-intrinsic roles for *Maf, Myo1f* and *Prr13*. A caveat of our system was that functional consequences of *Maf*-deficiency on antibody production or infection outcome could not be tested. Moreover, our system lacked any capacity for time-dependent gene-deletion. In the future, inducible Cre systems could be employed, for example, to determine the role of *Maf* solely during the later stages of infection or during secondary infection. A second outcome of studying *Maf* is new information on the function of this cMaf in experimental malaria. A previous study had shown that *Maf* is required for Tr1 cells to express the immune-suppressive cytokine, interleukin-10, in malaria [39]. In addition, we had previously confirmed PbTIIs express cMaf protein during peak *Pc*AS infection and retain this during transit to memory [11]. Given the dynamics of *Maf* expression in our model, along both Th1 and Tfh differentiation pathways, the emerging model is that cMaf is required for both Tfh and Tr1 differentiation in malaria. Future experiments are needed to determine possible roles for cMaf during repeated infections, which may be of importance given the frequency of infections experienced by children living in high transmission zones.

A broader question is why did predictions from transcriptional studies of *Myo1f* and *Prr13* not hold true when tested *in vivo*. The fact that mRNA species are variously subjected to post-transcriptional regulation, and may not necessarily lead to protein production offers one likely explanation. Nevertheless, the power of transcriptomic studies remains not only that they provide a broad overview of cellular processes, but ultimately, that assist in hypothesis generation. Recently, based on the same scRNA-seq data, we hypothesized and validated roles for IL-2 signalling and the chemokine receptor, CCR5, in balancing Thf/Tfh differentiation in malaria [38]. Hence it is clear that some upregulated genes function in CD4⁺ T cells, while others do not; empiricism continues to be required to determine this on a case by case basis. Thus, in summary, we have discovered that cMaf is critical in experimental malaria for the early differentiation of Tfh cells, while *Myo1f* and *Prr13* play no role in the biology of *Plasmodium*-specific CD4⁺ T cells during blood-stage infection.

## Methods

### Mice, genotyping, and phenotyping

**Mice.** *cMaf*-sufficient (*Maf* ⁽ᶠ/ᶠ⁾) PbTII and *cMaf*-deficient (CD4-Cre *Maf* ⁽ᶠ/ᶠ⁾) PbTII mice were generated, and protein expression of cMaf was confirmed by flow cytometry (Fig 2D). To obtain mice harbouring a PbTII cells-specific conditional knockout of either Prr13 or Myo1f driven by CD4Cre recombinase, *Prr13*- and *Myo1f*-deficient PbTII strain were generated as illustrated in Fig 3A-C. Prr3 flox mouse with loxP sites flanking exon 3 and 4 of the Prr13 gene and Myo1f flox mouse with loxP sites flanking exon 2 and 4 of the Myo1f gene (Fig 3A) were

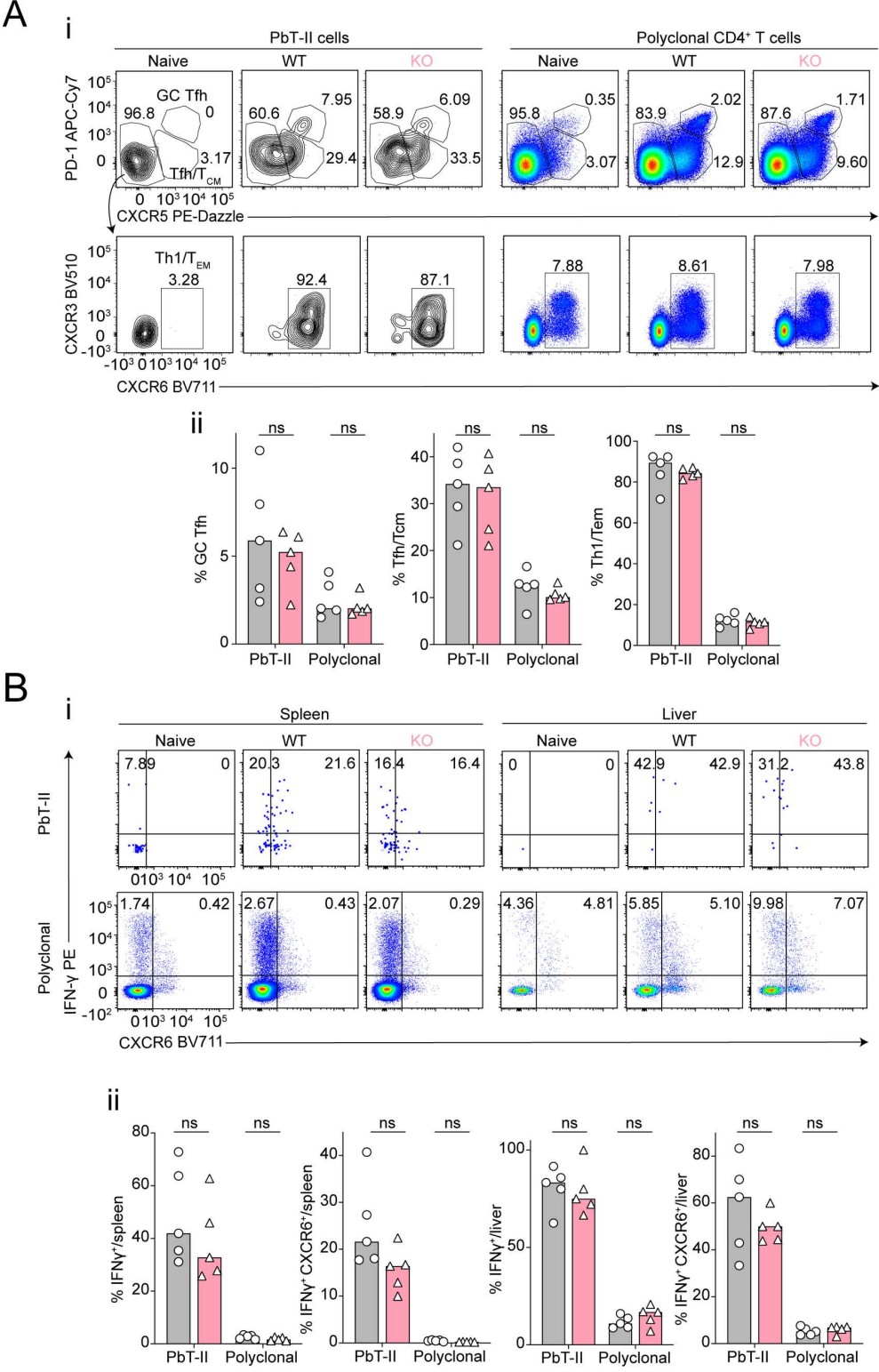

**Fig 5. *Prr13* does not control memory CD4+ T cell responses during experimental malaria.** (A & **B**) Spleno-cytes and hepatocytes were analysed at day 28 p.i. from the same experiment in Fig 4A. (**Ai**) Representative FACS plots of direct *ex vivo* CXCR5 and PD-1 expression gating CXCR5+PD-1+ GC Tfh and CXCR5+ Tfh/T_{CM} in PbTII and polyclonal CD4+ T cells in the *Prr13* [(f/f)] wild-type PbTIIs (WT) or CD4-Cre *Prr13* [(f/f)] PbTIIs (KO) group in the spleen at 28 d.p.i. The population of CXCR5- cells were further gated into CXCR3 and CXCR6 expression to define

CXCR5⁻CXCR3⁺CXCR6⁺ Th1/T$_{EM}$ cells. (Aii) Bar graphs showing frequencies of CXCR5⁺PD-1⁺ GC Tfh, CXCR5⁺ Tfh/T$_{CM}$, and CXCR5⁻CXCR3⁺CXCR6⁺ Th1/T$_{EM}$ cells. (Bi) Representative FACS plots of *ex vivo* stimulation-induced IFN-γ and CXCR6. (Bii) Bar graphs showing frequencies of IFN-γ⁺ or IFN-γ⁺ CXCR6⁺ PbTII and polyclonal CD4⁺ T cells in the WT and KO PbTII cells in the spleen and liver at day 28 p.i. (A & B) *n* = 5 per group, data are representative of two independent experiments. Statistical tests performed using unpaired, non-parametric Mann-Whitney t test; *$p$ < 0.05.

generated and purchased from Walter+Eliza Hall Institute of Medical Research (Melbourne, Victoria, Australia). Both CD4Cre and PbTII mouse were bred in-house. To generate knockout mice, PbTII mice and Prr13 or Myo1f flox mice were crossed in-house. Alongside, Prr13 or Myo1f flox mice and CD4Cre mice were crossed. Their offspring were further crossed to generate *Myo1f*-sufficient (*Myo1f* (f/f)) PbTII, *Myo1f*-deficient (CD4-Cre *Myo1f* (f/f)) PbTII, *Prr13*-sufficient (*Prr13* (f/f)) PbTII, and *Prr13*-deficient (CD4-Cre *Prr13* (f/f)) PbTII mice. For donor mice, Ly5.1 (CD45.1) wild-type C57BL/6J mice were obtained from Ozgene/ Animal Resources Centre (Western Australia). All animal experiments were conducted on female mice, aged 6-10 weeks, maintained under specific pathogen-free conditions at the Biological Research Facility of the Doherty Institute for Infection and Immunity (Melbourne, Victoria, Australia). Mice were monitored daily with criteria for humane euthanasia included severe weight loss (≥15% of initial body weight) and inability to eat or drink. Since anesthesia or analgesia was not required for our experimental protocol, all procedures were conducted to ensure minimal distress to the mice. All mice were euthanized via $CO_2$ exposure at day 7 or day 31 post-infection for tissue collection. The animal experiments, handling, and procedures were approved by the University of Melbourne Animal Ethics Committee (Ethics numbers: SLA-1: 1915018 & 10376).

**Genotyping.** Genotyping was performed from a small tissue biopsy of the ear using PCRs specific for the Cre transgene or the floxed region within the Prr13 and Myo1f gene. The following primers were used to detect Cre-wild type (wt) and transgene (Tg). Cre_wt_Fwd: CTAGGCCACAGAATTGAAAGATCT, Cre_wt_Rev: GTAGGTGGAAATTCTAGCATCATCC, Cre_Tg_ Fwd: GCGGTCTGGCAGTAAAAACTATC, and Cre_Tg_Rev: GTGAAACAGCATTGCTGTCACTT. The thermocycler for the Cre detection by PCR was performed by initial denaturation at 94°C for 2 minutes, annealing with 35 cycles of 94°C for 30 seconds, 55°C for 15 seconds, 72°C for 1 minute and finally extension at 72°C for 5 minutes.

For the detection of floxed regions of Prr13 and Myo1f, the following primers were used. Prr13_flox1_Fwd: CAGGGAGAGGGAAATGAGACTT, Prr13_flox1_Rev: TTTTCTTTC-CCGCATCCTCA, Prr13_flox2_Fwd: GTAACTTCCCTCACCCACCC, and Prr13_flox2_Rev: GGGCCATATGACCACTCAGT. Myo1f_flox1_Fwd: GGCCAAACTGACCTAGAGAG, Myo1f_flox1_Rev: TGAGTGCTGGGGTTAAAGGC, Myo1f_flox2_Fwd: ACCTGTA-ACTCTAGCATTGGGG, and Myo1f_flox2_Rev: CTGCCTTATTCCTTTGAGACAGG. The thermocycler for the Prr13 and Myo1f flox regions by PCR was performed by initial denaturation at 95°C for 2 minutes, annealing with 35 cycles of 95°C for 30 seconds, 58°C for Prr13 floxed or 60°C for Myo1f floxed for 15 seconds, 72°C for 30 seconds and finally extension at 72°C for 5 minutes. All primers were purchased from Sigma-Aldrich.

**Phenotyping.** To determine PbTII cell phenotype from the mice, the expression of CD4, TCRVα2, TCRVβ12 were analysed by flow cytometry. The drops of blood from the mice were lysed with Pharm Lyse (BD) and washed twice with PBS. Subsequently, cells were stained with the antibody cocktail constraining anti-CD4, anti-TCRVα2, and anti-TCRVβ12 antibody for

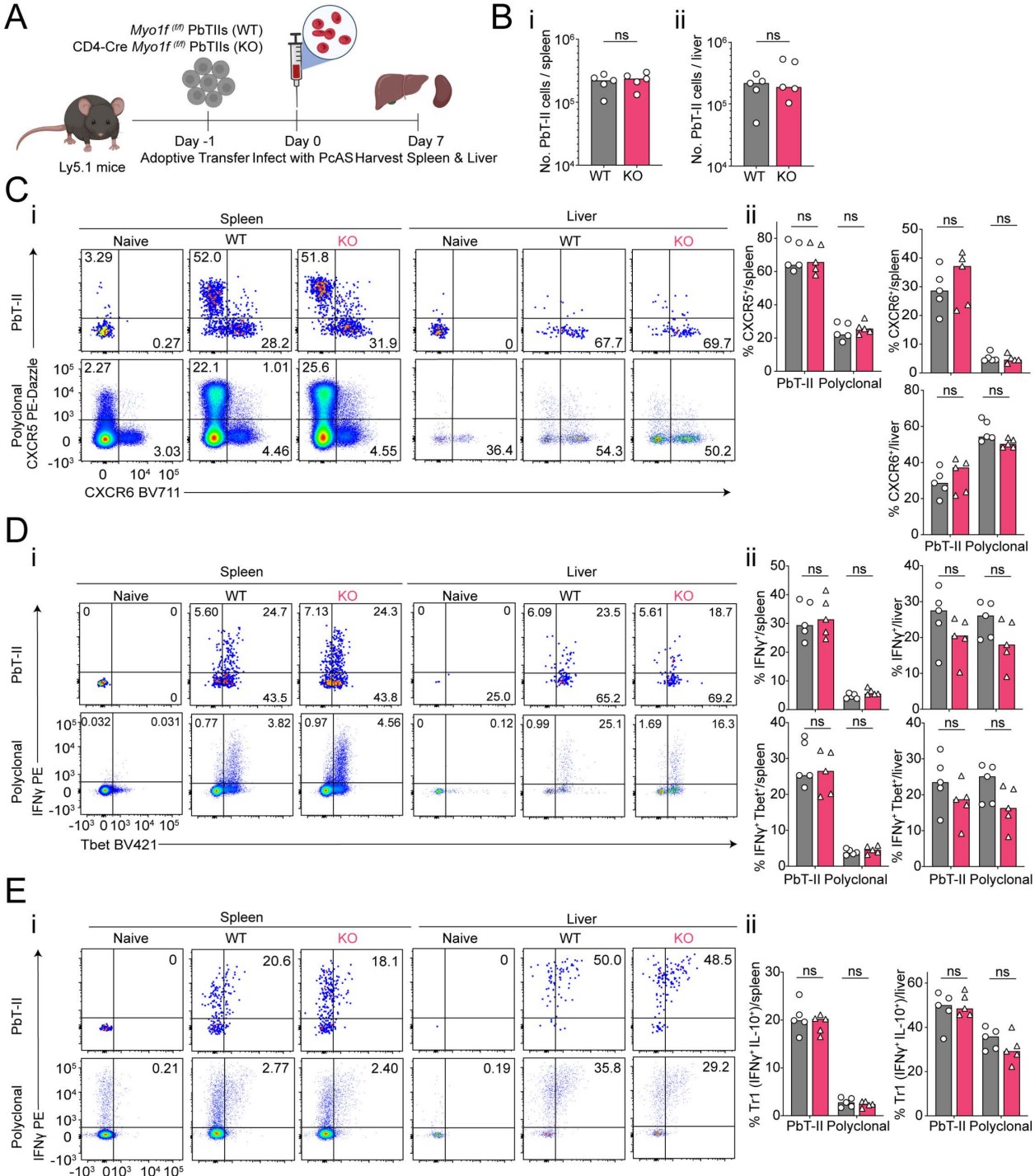

**Fig 6. *Myo1f* is dispensable for effector differentiation of *Plasmodium*-specific CD4 + T cells.** (A) Schematic of experiment: Congenically marked (CD45.2+) *Myo1f* (f/f) wild-type PbTIIs (WT) or CD4-Cre *Myo1f* (f/f) PbTIIs (KO) were adoptively transferred into Ly5.1 (CD45.1+) recipient mice infected with *Pc*AS infection a day 0. Splenocytes and liver cells were analysed on day 7 p.i. Created with Biorender.com. (**Bi & Bii**) Total number of PbTII cells per spleen and liver at day 7 p.i. (Ci) Representative FACS plots of CXCR5 and CXCR6 expression. (**Cii**) Bar graphs showing frequencies of CXCR5+ and/or CXCR6+ PbTII and polyclonal CD4+ T cells in the WT or KO group in the spleen and liver at day 7 p.i. (Di) Representative FASC plots of direct *ex vivo* IFN-γ and Tbet expression. (Dii) Bar graphs showing frequencies of IFN-γ+ or IFN-γ+ Tbet+ PbTII and polyclonal CD4+ T cells in WT or KO group in the spleen and liver at day 7 p.i. (Ei) Representative FACS plots of IFN-γ and IL-10

expression. (**Eii**) Bar graphs showing frequencies of IFN-γ⁺ IL-10⁺ Tr1 PbTII and polyclonal CD4⁺ T cells in the WT or KO group in the spleen and liver at day 7 p.i. Cells were stimulated with PMA/Ionomycin/Brefeldin A (BFA) for 3 hours at 37°C, 5% $CO_2$. (B-E) n = 5 per group, data are representative of two independent experiments. Statistical tests performed using unpaired, non-parametric Mann-Whitney t test; * $p < 0.05$.

30 minutes on ice in the dark. Following cell wash twice with PBS, the phenotype of cells was checked using Fortessa cytometer (BD). Data was analysed using FlowJo software (v.10.10).

## PbTII cell adoptive transfer, infection, and anti-malaria drug treatment

**PbTII cell adoptive transfer.** Naïve spleens from *cMaf*-sufficient (*Maf* ⁽ᶠ/ᶠ⁾) PbTII, *cMaf*-deficient (CD4-Cre *Maf* ⁽ᶠ/ᶠ⁾) PbTII, *Myo1f*-deficient (CD4-Cre *Myo1f* ⁽ᶠ/ᶠ⁾) PbTII, *Prr13*-sufficient (*Prr13* ⁽ᶠ/ᶠ⁾) PbTII, *Prr13*-deficient (CD4-Cre *Prr13* ⁽ᶠ/ᶠ⁾) PbTII mouse were harvested and homogenised through 70-100 μm cell strainers. The red blood cells (RBCs) were lysed using Pharm Lyse (BD) and CD4⁺ PbTII cells were enriched using CD4 microbeads (L3T4, Miltenyi Biotec). The puritywas determined by flow cytometry and ranged between 85% and 90%. Each donor mouse received $10^4$ PbTII cells via lateral tail vein injection.

**Infection.** Thawed PcAS infected blood from our biobank were used to infect C57BL/6J passage mice. *Pc*AS-infected RBCs (pRBCs) were obtained from the passage mice and processed to infect recipient mouse via lateral tail vein injection. $10^5$ pRBCs were injected for primary infection, whereas $10^7$ pRBCs were injected for re-infection.

**Parasitemia assessment.** A drop of blood was collected from the tail vein of mice into RPMI containing 5 U/mL of heparin. Diluted blood samples were stained with Hoechst 33342 (10 μg/mL; Sigma-Aldrich) and Syto84 (5 μM; Life Technologies) for 30 minutes at room temperature in the dark. The staining was quenched with 10 times the initial volume using cold RPMI. Immediately, the percentage of Syto84⁺ Hoechst33342⁺ RBCs was determined using Fortessa cytometer (BD Biosciences).

**Anti-malaria drug treatment.** Artesunate (Guilin Pharmaceutical, generously provided by J. Mohrle) was dissolved in a 5% sodium bicarbonate solution at a concentration of 50 mg/mL to create sodium artesunate and then diluted in 0.9% saline (Baxter) to reach a final concentration of 5 mg/mL. Mice were administered intraperitoneal injections of 1 mg of sodium artesunate twice daily from day 7 to day 9, once daily from day 10 to day 16, and twice weekly from day 17 to day 24 post-infection. Additionally, mice received pyrimethamine in their drinking water (70 mg/L, Sigma Aldrich) from day 7 until day 24 post-infection.

## Flow cytometry

Spleens were harvested into ice-cold RPMI and homogenized through 70/100μm cell strainers to create single-cell suspensions, followed by RBC lysis using Pharm Lyse (BD). Cells were then stained with Zombie Yellow viability dye (Biolegend) diluted in PBS, followed by Fc receptor block (BD). Finally, cells were stained with titrated panels of surface monoclonal antibodies (Table 1) diluted in PBS containing 2% FCS and 2 mM EDTA, where samples were incubated for 20 minutes on ice in the dark.

For intracellular staining, cells were incubated with brefeldin-A (10 mg/mL; Sigma) with or without PMA (1 mg/mL; Sigma) and ionomycin (1 mg/mL; Sigma) at 37°C for 3h. After incubation, Foxp3/ Transcription Factor Staining Buffer Set (eBiosciences) or BD Cytofix/ Cytoperm Fixation/Permeabilization Kit (BD) was used to fix and permeabilise cells prior to staining with panels of monoclonal antibodies (Table 1) for 30-90 minutes on ice in the dark. Finally, cells were washed and acquired on Fortessa cytometer (BD). The gating strategy was indicated in Fig 2C, and all data was analysed using FlowJo (v.10.10).

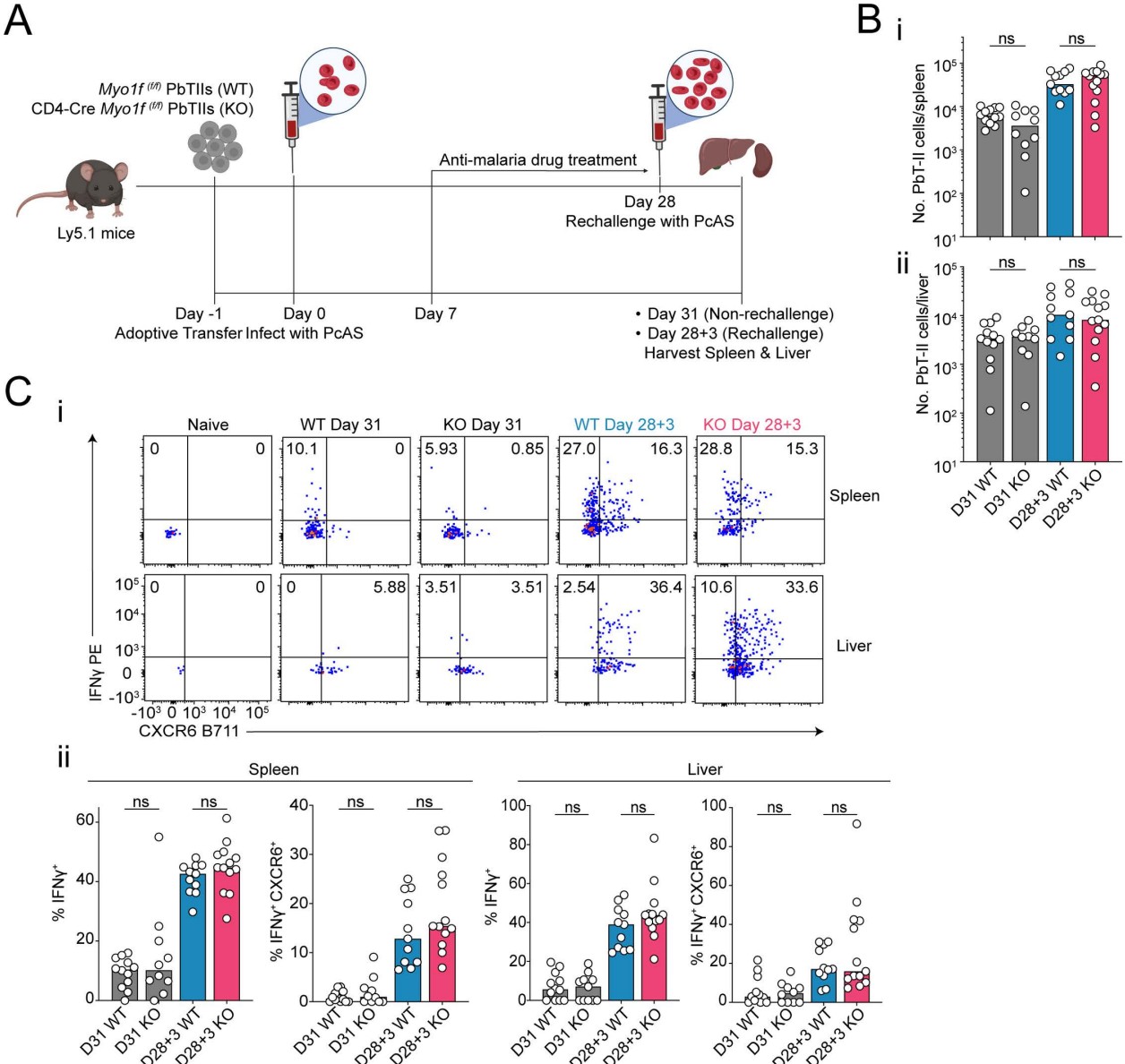

**Fig 7. *Myo1f* is dispensable for *Plasmodium*-specific CD4$^+$ T cells during *Pc*AS rechallenge.** (A) Schematic of experiment: congenically marked (CD45.2$^+$) *Myo1f* $^{(f/f)}$ wild-type PbTIIs (WT) or CD4-Cre *Myo1f* $^{(f/f)}$ PbTIIs (KO) were adoptively transferred into Ly5.1 (CD45.1$^+$) recipient mice infected with *Pc*AS infection a day 0. Mice were under the anti-malaria drug treatment from day 7 p.i. Mice were either rechallenged or non-rechallenged at day 28 p.i. 3 days later, the spleen and liver were harvested from rechallenge mice (Day 28 + 3) and non-rechallenge mice (Day 31). Created with Biorender.com. (**Bi & Bii**) Total number of PbTIIs per spleen and liver at day 28 + 3/day 31 p.i. (Ci) Representative FACS plots of direct *ex vivo* IFN-γ and CXCR6. (Cii) Bar graphs showing frequencies of IFN-γ$^+$ or IFN-γ$^+$ CXCR6$^+$ in WT or KO PbTII cells in the spleen and liver at day 28 + 3/day 31 p.i. (B & C) *n* = 5-7 per group, data are pooled from two independent experiments. Statistical test performed using paired two-way ANOVA with Tukey's multiple comparison test; * *p* < 0.05.

## Statistical analysis

Statistical analyses were performed using Prism (v10.2.3 GraphPad Software, Boston, Massachusetts, USA) and tests were performed using either an unpaired, non-parametric Mann-Whitney t test or paired two-way ANOVA with Tukey's multiple comparison test. *P* value < 0.05 was considered significant.

**Table 1. The list of surface and intracellular antibody.**

| Surface antibody | Fluorophore | Clone | Source | Titration |
|---|---|---|---|---|
| CD4 | AF700 | RM4-5 | BioLegend | 1:200 |
| CXCR3 | BV510 | CXCR3-173 | BioLegend | 1:200 |
| CXCR5 | Biotin | 2G8 | BD | 1:50 |
| SAV | BV786 | | BioLegend | 1:200 |
| CXCR5 | PE-Dazzle594 | L138D7 | BioLegend | 1:200 |
| CXCR6 | BV711 | SA051D1 | BioLegend | 1:200 |
| CD11a | PerCP-Cy5.5 | M17/4 | BioLegend | 1:200 |
| CD45.2 | FITC | 104 | BioLegend | 1:200 |
| TCRβ | BUV737 | H57-5797 | BD | 1:200 |
| TCRVα2 | APC-Cy7 | B20.1 | BioLegend | 1:200 |
| **Intracellular antibody** | **Fluorophore** | **Clone** | **Source** | **Titration** |
| CCL5 | PE-Cy7 | 2E9 | BioLegend | 1:200 |
| IFN-γ | PE | XMG1.2 | BioLegend | 1:200 |
| IL-10 | AF-647 | JES5-16E3 | BioLegend | 1:100 |
| Bcl6 | PE | K112-91 | BioLegend | 1:50 |
| Tbet | BV421 | 4B10 | BioLegend | 1:200 |
| cMaf | PE | T54-853 | BD | 1:200 |

## Supporting information

**S1 Fig. Raw gel images.**
(TIF)

## Author contributions

**Conceptualization:** Takahiro Asatsuma, Marcela L. Moreira, Ashraful Haque.

**Data curation:** Takahiro Asatsuma, Marcela L. Moreira.

**Formal analysis:** Takahiro Asatsuma, Marcela L. Moreira, Hyun J. Lee, Cameron G. Williams.

**Funding acquisition:** Ashraful Haque.

**Investigation:** Takahiro Asatsuma, Marcela L. Moreira, Hyun J. Lee, Oliver P. Skinner, Shihan Li, Ivana Rea, Taidhgin Harkin, Saba Asad.

**Methodology:** Takahiro Asatsuma, Marcela L. Moreira.

**Project administration:** Marcela L. Moreira, Ashraful Haque.

**Supervision:** Lynette Beattie, Ashraful Haque.

**Validation:** Takahiro Asatsuma.

**Visualization:** Takahiro Asatsuma.

**Writing – original draft:** Takahiro Asatsuma, Marcela L. Moreira.

**Writing – review & editing:** Brooke J. Wanrooy, Ashraful Haque.

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
