## [Decision Letter · Decision Letter 0]

23 Oct 2024

PONE-D-24-38309Myosin 1f and Proline-rich 13 are transcriptionally upregulated yet functionally redundant in T helper cells during experimental malaria.PLOS ONE

Dear Dr. Haque,

Your manuscript was reviewed by two experts in the field and both of them were enthusiastic about the findings reported in your manuscript. There were, however, some concerns that were raised by one of the reviewers that need to be addressed prior to making a final decision on your manuscript. As such, my recommendation is minor revision. Therefore, we invite you to submit a revised version of the manuscript that addresses the points raised during the review process. Please submit your revised manuscript by Dec 07 2024 11:59PM. If you will need more time than this to complete your revisions, please reply to this message or contact the journal office at plosone@plos.org . Please include the following items when submitting your revised manuscript:

We look forward to receiving your revised manuscript.

Kind regards,

Dr. Joseph J Mattapallil

Academic Editor

PLOS ONE

Journal Requirements:

2. To comply with PLOS ONE submissions requirements, in your Methods section, please provide additional information regarding the experiments involving animals and ensure you have included details on methods of anesthesia and/or analgesia.

3. We note that Figure 1 in your submission contain map/satellite images which may be copyrighted. All PLOS content is published under the Creative Commons Attribution License (CC BY 4.0), which means that the manuscript, images, and Supporting Information files will be freely available online, and any third party is permitted to access, download, copy, distribute, and use these materials in any way, even commercially, with proper attribution. For these reasons, we cannot publish previously copyrighted maps or satellite images created using proprietary data, such as Google software (Google Maps, Street View, and Earth). For more information, see our copyright guidelines: http://journals.plos.org/plosone/s/licenses-and-copyright. We require you to either (a) present written permission from the copyright holder to publish these figures specifically under the CC BY 4.0 license, or (b) remove the figures from your submission:

a. You may seek permission from the original copyright holder of Figure 1 to publish the content specifically under the CC BY 4.0 license. We recommend that you contact the original copyright holder with the Content Permission Form (http://journals.plos.org/plosone/s/file?id=7c09/content-permission-form.pdf) and the following text: “I request permission for the open-access journal PLOS ONE to publish XXX under the Creative Commons Attribution License (CCAL) CC BY 4.0 (http://creativecommons.org/licenses/by/4.0/). Please be aware that this license allows unrestricted use and distribution, even commercially, by third parties. Please reply and provide explicit written permission to publish XXX under a CC BY license and complete the attached form.” Please upload the completed Content Permission Form or other proof of granted permissions as an "Other" file with your submission. In the figure caption of the copyrighted figure, please include the following text: “Reprinted from [ref] under a CC BY license, with permission from [name of publisher], original copyright [original copyright year].”

b. If you are unable to obtain permission from the original copyright holder to publish these figures under the CC BY 4.0 license or if the copyright holder’s requirements are incompatible with the CC BY 4.0 license, please either i) remove the figure or ii) supply a replacement figure that complies with the CC BY 4.0 license. Please check copyright information on all replacement figures and update the figure caption with source information. If applicable, please specify in the figure caption text when a figure is similar but not identical to the original image and is therefore for illustrative purposes only. The following resources for replacing copyrighted map figures may be helpful: USGS National Map Viewer (public domain): http://viewer.nationalmap.gov/viewer/ The Gateway to Astronaut Photography of Earth (public domain): http://eol.jsc.nasa.gov/sseop/clickmap/ Maps at the CIA (public domain): https://www.cia.gov/library/publications/the-world-factbook/index.html and https://www.cia.gov/library/publications/cia-maps-publications/index.html NASA Earth Observatory (public domain): http://earthobservatory.nasa.gov/ Landsat: http://landsat.visibleearth.nasa.gov/ USGS EROS (Earth Resources Observatory and Science (EROS) Center) (public domain): http://eros.usgs.gov/# Natural Earth (public domain): http://www.naturalearthdata.com/

4. Please upload a copy of Supporting Information Figure 1 which you refer to in your text on page 18.

5. PLOS ONE now requires that authors provide the original uncropped and unadjusted images underlying all blot or gel results reported in a submission’s figures or Supporting Information files. This policy and the journal’s other requirements for blot/gel reporting and figure preparation are described in detail at https://journals.plos.org/plosone/s/figures#loc-blot-and-gel-reporting-requirements and https://journals.plos.org/plosone/s/figures#loc-preparing-figures-from-image-files. When you submit your revised manuscript, please ensure that your figures adhere fully to these guidelines and provide the original underlying images for all blot or gel data reported in your submission. See the following link for instructions on providing the original image data: https://journals.plos.org/plosone/s/figures#loc-original-images-for-blots-and-gels. In your cover letter, please note whether your blot/gel image data are in Supporting Information or posted at a public data repository, provide the repository URL if relevant, and provide specific details as to which raw blot/gel images, if any, are not available. Email us at plosone@plos.org if you have any questions.

Reviewers' comments:

Reviewer's Responses to Questions

**Comments to the Author**

1. Is the manuscript technically sound, and do the data support the conclusions?

Reviewer #1: Partly

Reviewer #2: Yes

2. Has the statistical analysis been performed appropriately and rigorously? 

Reviewer #1: Yes

Reviewer #2: Yes

3. Have the authors made all data underlying the findings in their manuscript fully available?

Reviewer #1: No

Reviewer #2: Yes

4. Is the manuscript presented in an intelligible fashion and written in standard English?

Reviewer #1: Yes

Reviewer #2: Yes

5. Review Comments to the Author

Reviewer #1: Positive feedback. I applaud the authors for moving forward with a publication of negative results. It is important to show that some suspected genes are not important in T cell responses (to malaria). The text is well written, figures are of a good quality (but see comments), and results seems solid (given performed experiments, see also below). I also appreciated authors showing the data from individual mice; this is important and should be the standard in the field.

Comments.

1. The rationalization of choosing these two genes to study is still poor. Ok, these genes are unregulated in scRNAseq data. So what? I predict hundreds of genes are unregulated - why these two? Are they unregulated to the highest level (mRNA)? Are they upregulated at the protein level? I found the data presented in Fig 1A being unclear. NOTE: because individual panels are not labeled, it is hard to refer to individual panels. But, in Fig 1A trajectories - how uncertain are these? In Fig 1A violin plots -> what are these (no labels given)? No stats are given either.

2. Measurements are mostly done at day 7. Why? Could it be too early to see a difference? Is there a positive control authors could present that would justify measurements at day 7 (or 28) as appropriate?

3. I could not understand the results shown in Fig 3D - the way I see these gels is that expression in WT vs. KO is the same, i.e., KO did not work. I think someone experienced in gels should look into these (e.g., Dr Elisabeth Bik @MicrobiomDigest on Twitter)

4. I am not sure about some statistical analyses showing no difference between WT and KO cells. For example, in Fig 4B and 4D (cannot point to panels!), there is a tendency of lower total T cell numbers in PbT-II KO cells. I wonder if experiments in which WT and KO cells are co-transferred to the mice and then look at the response magnitude. This way, the data will be paired and may allow for higher statistical power to detect difference.

Minor comments.

Improvements in figures must be made. Every panel in a figure should be "named", e.g., you can use Ai, Aii, Aiii, etc to label sub-panels in a figure with multiple elements (A/B/C etc).

I highly recommend generating pdf of the text with ALL figures and figure captions being on the same page, e.g., as done in published papers or grant applications. This makes reviewing of the paper easier. Also, number the lines so reference to specific lines in the text can be made. And obvious -> have page numbers!

The text in 2nd paragraph in Results is unclear. It reads as you put T cells into WT Cre-expressing mice but it does not seem to be correct as you generated KO mice first. Or perhaps I did not understand the text well -> in any case, please review and update if needed.

If KO T cells generate fewer Tfh cells, does this result in lower Ab (IgG?) levels to the parasites? Perhaps this needs to be shown.

The quality of the KO is stated in Methods to be shown in Fig 2B (cannot point more precisely because page and line numbers are missing). I cannot see this. Please make sure you refer to the right figures/panels.

Purity of CD4 T cells was exceeding approximately 80%. What does this mean? 81%? Or 99%. Please put specific numbers/range, e.g., 80-85%.

Statistical comparisons are not properly presented. In particular, it is not recommended anymore to use stars (**) but putting exact p values, the same for "ns" -> put exact value. Also, if possible, indicating fold change is highly recommended as showing effect size is as important as showing p values.

Given variability in some data (indicated by measurements in individual mice) - e.g., Fig 5A, how much does a gene deletion must impact Tfh (or something else) differentiation, so differences become statistically significant? Seems that with such variability these experiments may be simply underpowered to detect a difference. Experiments in which measurements are "paired" may be thus more powerful (see a suggestion above). This may need to be discussed.

Reviewer #2: This study carefully evaluates the potential role of Maf1, Myo1f and Prr13 on CD4 T cell maturation following Plasmodium chabaudi chabaudi AS (PcAS) infections in mice. All three genes had previously been shown to be upregulated during T cell differentiation in response to PcAS. They assessed both Plasmodium specific-TCR transgenic and polyclonal T cells and found the anticipated decrease in Tfh cell production following Maf1 deletion in Transgenic CD4+ cells, but neither Myo1f nor Prr13 deletion had a measurable effect. Myo1f or Prr13 have not been tested before for a role in T cell activity so this well-designed study advances the understanding of genes that although upregulated are not critical for CD4 T cell differentiation in response to malaria.

6. PLOS authors have the option to publish the peer review history of their article (what does this mean? ). If published, this will include your full peer review and any attached files.

**Do you want your identity to be public for this peer review?** For information about this choice, including consent withdrawal, please see our Privacy Policy .

Reviewer #1: No

Reviewer #2: No

---

## [Author Response · Author response to Decision Letter 0]

15 Dec 2024

Rebuttal Letter

PLOS ONE PONE-D-24-38309

Myosin 1f and Proline-rich 13 are transcriptionally upregulated yet functionally redundant in T helper cells during experimental malaria.

Editor’s comments

Dear Dr. Haque,

Your manuscript was reviewed by two experts in the field and both of them were enthusiastic about the findings reported in your manuscript. There were, however, some concerns that were raised by one of the reviewers that need to be addressed prior to making a final decision on your manuscript. As such, my recommendation is minor revision. Therefore, we invite you to submit a revised version of the manuscript that addresses the points raised during the review process.

We look forward to receiving your revised manuscript.

Kind regards,

Dr. Joseph J Mattapallil

Academic Editor

PLOS ONE

We thank the editor and reviewers for their comments and suggestions on our manuscript. We have thoroughly addressed all concerns. Below, we provide a point-by-point response to the reviewers’ comments, with our responses highlighted in BLUE text. Where appropriate, we have included new text that have been incorporated into the revised manuscript, along with line numbers for ease of reference.

Journal Requirements:

We have revised the manuscript to follow PLOS ONE's style requirements

2. To comply with PLOS ONE submissions requirements, in your Methods section, please provide additional information regarding the experiments involving animals and ensure you have included details on methods of anesthesia and/or analgesia.

In the revised Methods section, we have now provided additional information regarding animal care and welfare as follows (line 378): “All animal experiments were conducted on female mice....” Specifically, we have included details on the monitoring criteria used for humane endpoints, the method of euthanasia, and the approvals obtained from the University of Melbourne Animal Ethics Committee. As anesthesia or analgesia was not required for our experimental protocol, we have specified that all procedures were conducted with minimal distress to the mice.

3. We note that Figure 1 in your submission contain map/satellite images which may be copyrighted. All PLOS content is published under the Creative Commons Attribution License (CC BY 4.0), which means that the manuscript, images, and Supporting Information files will be freely available online, and any third party is permitted to access, download, copy, distribute, and use these materials in any way, even commercially, with proper attribution. For these reasons, we cannot publish previously copyrighted maps or satellite images created using proprietary data, such as Google software (Google Maps, Street View, and Earth). For more information, see our copyright guidelines: http://journals.plos.org/plosone/s/licenses-and-copyright. We require you to either (a) present written permission from the copyright holder to publish these figures specifically under the CC BY 4.0 license, or (b) remove the figures from your submission:

a. You may seek permission from the original copyright holder of Figure 1 to publish the content specifically under the CC BY 4.0 license. We recommend that you contact the original copyright holder with the Content Permission Form (http://journals.plos.org/plosone/s/file?id=7c09/content-permission-form.pdf) and the following text: “I request permission for the open-access journal PLOS ONE to publish XXX under the Creative Commons Attribution License (CCAL) CC BY 4.0 (http://creativecommons.org/licenses/by/4.0/). Please be aware that this license allows unrestricted use and distribution, even commercially, by third parties. Please reply and provide explicit written permission to publish XXX under a CC BY license and complete the attached form.” Please upload the completed Content Permission Form or other proof of granted permissions as an "Other" file with your submission. In the figure caption of the copyrighted figure, please include the following text: “Reprinted from [ref] under a CC BY license, with permission from [name of publisher], original copyright [original copyright year].”

b. If you are unable to obtain permission from the original copyright holder to publish these figures under the CC BY 4.0 license or if the copyright holder’s requirements are incompatible with the CC BY 4.0 license, please either i) remove the figure or ii) supply a replacement figure that complies with the CC BY 4.0 license. Please check copyright information on all replacement figures and update the figure caption with source information. If applicable, please specify in the figure caption text when a figure is similar but not identical to the original image and is therefore for illustrative purposes only. The following resources for replacing copyrighted map figures may be helpful: USGS National Map Viewer (public domain): http://viewer.nationalmap.gov/viewer/ The Gateway to Astronaut Photography of Earth (public domain): http://eol.jsc.nasa.gov/sseop/clickmap/ Maps at the CIA (public domain): https://www.cia.gov/library/publications/the-world-factbook/index.html and https://www.cia.gov/library/publications/cia-maps-publications/index.html NASA Earth Observatory (public domain): http://earthobservatory.nasa.gov/ Landsat: http://landsat.visibleearth.nasa.gov/ USGS EROS (Earth Resources Observatory and Science (EROS) Center) (public domain): http://eros.usgs.gov/# Natural Earth (public domain): http://www.naturalearthdata.com/

We previously consulted Dr. Nick Bernard, a Senior Editor at Nature Immunology regarding the reuse of figures from the study by Soon et al. (ref. 11) in publications with Cell Reports (ref. 38) and Nature Communications (ref. 15). In those instances, they confirmed that providing a citation was sufficient, with no additional permissions or paperwork required. The permission statements in the figure legend of Figure. 1 as follows (line 138); “(A-D) Reprinted from....”

4. Please upload a copy of Supporting Information Figure 1 which you refer to in your text on page 18.

Thank you for pointing this out, and we apologize for any confusion. This was an error in the original manuscript. The reference to "Supporting Information Figure 1" was intended to refer to Figure 2C, as we decided not to include supplementary figures. We have corrected this in the revised manuscript for clarity (line 367)

5. PLOS ONE now requires that authors provide the original uncropped and unadjusted images underlying all blot or gel results reported in a submission’s figures or Supporting Information files. This policy and the journal’s other requirements for blot/gel reporting and figure preparation are described in detail at https://journals.plos.org/plosone/s/figures#loc-blot-and-gel-reporting-requirements and https://journals.plos.org/plosone/s/figures#loc-preparing-figures-from-image-files. When you submit your revised manuscript, please ensure that your figures adhere fully to these guidelines and provide the original underlying images for all blot or gel data reported in your submission. See the following link for instructions on providing the original image data: https://journals.plos.org/plosone/s/figures#loc-original-images-for-blots-and-gels. In your cover letter, please note whether your blot/gel image data are in Supporting Information or posted at a public data repository, provide the repository URL if relevant, and provide specific details as to which raw blot/gel images, if any, are not available. Email us at plosone@plos.org if you have any questions.

We have submitted the original, uncropped, and unadjusted gel images as the S1_raw_image file alongside the revised manuscript. In addition, the new Figure 3C in the revised version now retains the original gel results with minimal cropping and adjustment, ensuring transparency and preserving the integrity of the data.

Thank you for the reminder. We have reviewed the reference list to ensure its accuracy.

Reviewer's Responses to Questions

1. Is the manuscript technically sound, and do the data support the conclusions?

Reviewer #1: Partly

Reviewer #2: Yes

2. Has the statistical analysis been performed appropriately and rigorously?

Reviewer #1: Yes

Reviewer #2: Yes

3. Have the authors made all data underlying the findings in their manuscript fully available?

Reviewer #1: No

Reviewer #2: Yes

4. Is the manuscript presented in an intelligible fashion and written in standard English?

Reviewer #1: Yes

Reviewer #2: Yes

5. Reviewer's Comments to the Author

Reviewer #1:

Positive feedback. I applaud the authors for moving forward with a publication of negative results. It is important to show that some suspected genes are not important in T cell responses (to malaria). The text is well written, figures are of a good quality (but see comments), and results seems solid (given performed experiments, see also below). I also appreciated authors showing the data from individual mice; this is important and should be the standard in the field.

Comments.

1. The rationalization of choosing these two genes to study is still poor. Ok, these genes are unregulated in scRNAseq data. So what? I predict hundreds of genes are unregulated - why these two? Are they unregulated to the highest level (mRNA)? Are they upregulated at the protein level? I found the data presented in Fig 1A being unclear. NOTE: because individual panels are not labeled, it is hard to refer to individual panels. But, in Fig 1A trajectories - how uncertain are these? In Fig 1A violin plots -> what are these (no labels given)? No stats are given either.

Thank you for your question regarding the rationale behind our selection of Myo1f and Prr13 for this study. Our choice was driven by both the strong upregulation of these genes in our scRNA-seq dataset and functional insights from prior studies, which pointed to their potential roles in T cell biology during infection.

For Myo1f, previous studies have demonstrated its role in controlling cell motility and granule exocytosis in neutrophils and other innate immune cells (ref. 21, 24-27). This evidence led us to hypothesize that Myo1f might similarly regulate T cell movement and function during infection. Its consistent high expression specifically in effector and memory CD4+ T cells during blood-stage malaria suggested that Myo1f could influence T cell movement and/or function in experimental malaria, making it a compelling candidate for further investigation. We have addressed this point by revising the text as follows (line 111): “Given prior evidence of Myo1f’s role....”

For Prr13, sequence analyses (UniProt and NCBI) and prior studies (ref. 31, 33-35) indicate that it may encode a DNA-binding protein localized in the nucleus with transcriptional regulatory functions. This evidence led us to hypothesize that Prr13 might act as a transcriptional factor in T cells, potentially controlling differentiation and cellular maintenance. Thus, the distinct and elevated expression of Prr13 in effector and memory T cells pointed toward a possible role in T cell fate decisions and memory maintenance in experimental blood-stage malaria. We have addressed this point by revising the text as follows (line 118): “Given sequence predictions and prior studies suggesting that Prr13....” We hope this response clarifies our rationale and highlights the potential of Myo1f and Prr13 to provide new insights into T cell function and differentiation in the context of infection.

We apologize for the lack of labelling and clarity in Fig. 1A. In the revised figure, we have now sub-labelled each panel within Fig. 1A. The violin plots represent mRNA expression levels in PbTIIs across different time points (day 0, day 7, day 14, and day 28 post-infection) which helps identify the gene expression dynamics of Myo1f and Prr13 during PcAS infection. Also, we considered performing single-cell level statistics. However, this approach often results in inflated p-values, as each cell is treated as an individual sample, potentially making all comparisons appear statistically significant without reflecting biologically meaningful differences. Therefore, to provide an accurate representation of our findings, we have chosen not to perform statistical tests at t

---

## [Decision Letter · Decision Letter 1]

17 Jan 2025

PONE-D-24-38309R1Myosin 1f and Proline-rich 13 are transcriptionally upregulated yet functionally redundant in CD4+ T cells during blood-stage Plasmodium infection.PLOS ONE

Dear Dr. Haque,

Thank you for submitting your manuscript to PLOS ONE. After careful consideration, we feel that it has merit but does not fully meet PLOS ONE’s publication criteria as it currently stands. Therefore, we invite you to submit a revised version of the manuscript that addresses the points raised during the review process.

We look forward to receiving your revised manuscript.

Kind regards,

Joseph J Mattapallil

Academic Editor

PLOS ONE

Journal Requirements:

Reviewers' comments:

Reviewer's Responses to Questions

**Comments to the Author**

1. If the authors have adequately addressed your comments raised in a previous round of review and you feel that this manuscript is now acceptable for publication, you may indicate that here to bypass the “Comments to the Author” section, enter your conflict of interest statement in the “Confidential to Editor” section, and submit your "Accept" recommendation.

Reviewer #1: (No Response)

Reviewer #2: All comments have been addressed

2. Is the manuscript technically sound, and do the data support the conclusions?

Reviewer #1: Partly

Reviewer #2: Yes

3. Has the statistical analysis been performed appropriately and rigorously? 

Reviewer #1: I Don't Know

Reviewer #2: Yes

4. Have the authors made all data underlying the findings in their manuscript fully available?

Reviewer #1: No

Reviewer #2: Yes

5. Is the manuscript presented in an intelligible fashion and written in standard English?

Reviewer #1: Yes

Reviewer #2: Yes

6. Review Comments to the Author

Reviewer #1: Comments.

1. The authors provided some justification of why these 2 genes (Myo1f and Prr13) were perhaps selected for further analysis but their abstract and most of text still did not explicitly say that prior knowledge was used. In particular, abstract ignores this completely. Please revised and put explicit statements that selection of these 2 genes was also driven by prior knowledge of importance of these genes in T cell responses and not just by scRNAseq data (which give hundreds of unregulated genes).

2. I understand your points of choosing day 7 to measure responses but I think you do not state that clearly in the main text. Please do.

3. The study could be underpowered - as authors agreed, so this should be listed in the Discussion, along with the point that you could not do experiments when co-transferring WT and KO cells due to cogenic marker identity.

4. I appreciate showing original gels but I still could not understand them. It could be my limitation, so editor may need to show these to the right expert to ensure they are accurate.

Minor comments

You did not format the figures/captions properly as we do in grants (perhaps you don't apply for funding). You MUST show the figure and caption together! (And preferably on the page near where that figure is first cited). I will not review another paper that is not formatted to help easier review process.

If Figure 1 has been published before (per disclaimer listed), why do we need it here? The authors anyway state that they focused on the genes found by mining previously published dataset (along with some prior knowledge), so do we need to see Figure 1? NOTE: not all panels in Figure 1 are labeled, e.g., Violin plots in A are not. And, you may want to mention that only about 50% of cells (based on shape of violin plots) express Myo1f (the middle panel in Figure 1A) -> is that important that 50% of cells do not upregulate this gene?

Please remove "ns" from all panels and put actual p values there. You did not interpret my comment correctly - do not use shortcuts but put actual p values, i.e., p=0.89 if difference is statistically non-significant.

The experiments with co-transfer of WT and KO cells could still perhaps be done (in the future) by using some type of cell label, e.g., CTV. The label does get diluted but CD4s are not expected to divide that much, so some level of labeling may be retained by day 7. Perhaps it is worth mentioning as future directions in the paper.

Line 113: do you mean "our data" or "our analysis"? Data had been published.

Line 139: "infected with PcAS infection" sounds weird. Buttery butter.

Line 145: you must include justification of why day 7 is a good day to measure this response.

Line 311: include justification of why these genes are good ones by citing relevant other studies.

340: "regulated" -> unregulated (?)

Reviewer #2: The rationale and figures have been clarified and the authors have responded to all the criticisms within the limits of the genetics of the experimental design.

7. PLOS authors have the option to publish the peer review history of their article (what does this mean? ). If published, this will include your full peer review and any attached files.

**Do you want your identity to be public for this peer review?** For information about this choice, including consent withdrawal, please see our Privacy Policy .

Reviewer #1: No

Reviewer #2: No

---

## [Author Response · Author response to Decision Letter 1]

4 Feb 2025

All minor changes requested by one of the Reviewers have been adopted, and are detailed in the "Response to Reviewer" file

---

## [Decision Letter · Decision Letter 2]

18 Feb 2025

Myosin 1f and Proline-rich 13 are transcriptionally upregulated yet functionally redundant in CD4+ T cells during blood-stage Plasmodium infection.

PONE-D-24-38309R2

Dear Dr. Haque,

We’re pleased to inform you that your manuscript has been judged scientifically suitable for publication and will be formally accepted for publication once it meets all outstanding technical requirements.

Kind regards,

Joseph J Mattapallil

Academic Editor

PLOS ONE

Additional Editor Comments (optional):

Reviewers' comments:

Reviewer's Responses to Questions

**Comments to the Author**

1. If the authors have adequately addressed your comments raised in a previous round of review and you feel that this manuscript is now acceptable for publication, you may indicate that here to bypass the “Comments to the Author” section, enter your conflict of interest statement in the “Confidential to Editor” section, and submit your "Accept" recommendation.

Reviewer #1: All comments have been addressed

2. Is the manuscript technically sound, and do the data support the conclusions?

Reviewer #1: Yes

3. Has the statistical analysis been performed appropriately and rigorously? 

Reviewer #1: Yes

4. Have the authors made all data underlying the findings in their manuscript fully available?

Reviewer #1: No

5. Is the manuscript presented in an intelligible fashion and written in standard English?

Reviewer #1: Yes

6. Review Comments to the Author

Reviewer #1: I don't see that you fully addressed my comments but because they are becoming a mundane point, I give up. Why you chose those two genes remains a puzzle to me - but again, perhaps it could remain a puzzle.

7. PLOS authors have the option to publish the peer review history of their article (what does this mean? ). If published, this will include your full peer review and any attached files.

**Do you want your identity to be public for this peer review?** For information about this choice, including consent withdrawal, please see our Privacy Policy .

Reviewer #1: No

---

## [Editor Report · Acceptance letter]

PONE-D-24-38309R2

PLOS ONE

Dear Dr. Haque,

I'm pleased to inform you that your manuscript has been deemed suitable for publication in PLOS ONE. Congratulations! Your manuscript is now being handed over to our production team.

Kind regards,

on behalf of

Dr. Joseph J Mattapallil

Academic Editor

PLOS ONE